



**Distinctions in source regions and formation mechanisms of secondary aerosol in Beijing from summer to winter**

Jing Duan[1,2,3], Ru-Jin Huang[1,2], Chunshui Lin[1,2,4], Wenting Dai[1,2], Meng Wang[1,2,3], Yifang Gu[1,2,3], Ying Wang[1,2,3], Haobin Zhong[1,2,3], Yan Zheng[5], Haiyan Ni[1,2,3,6], Uli Dusek[6], Yang Chen[7], Yongjie Li[8], Qi Chen[5], Douglas R. Worsnop[9], Colin D. O'Dowd[4], Junji Cao[1,2]

[1]State Key Laboratory of Loess and Quaternary Geology (SKLLQG) and Key Laboratory of Aerosol Chemistry & Physics (KLACP), Institute of Earth Environment, Chinese Academy of Sciences, Xi'an 710061, China
[2]CAS Center for Excellence in Quaternary Science and Global Change, Chinese Academy of Sciences, Xi'an 710061, China
[3]University of Chinese Academy of Sciences, Beijing 100049, China
[4]School of Physics and Centre for Climate and Air Pollution Studies, Ryan Institute, National University of Ireland Galway, University Road, Galway, Ireland
[5]State Key Joint Laboratory of Environmental Simulation and Pollution Control, College of Environmental Sciences and Engineering, Peking University, Beijing 100871, China
[6]Centre for Isotope Research (CIO), Energy and Sustainability Research Institute Groningen (ESRIG), University of Groningen, The Netherlands
[7]Chongqing Institute of Green and Intelligent Technology, Chinese Academy of Sciences, Chongqing 400714, China
[8]Department of Civil and Environmental Engineering, Faculty of Science and Technology, University of Macau, Taipa, Macau
[9]Aerodyne Research, Inc., Billerica, MA, USA

*Correspondence to*: Ru-Jin Huang (rujin.huang@ieecas.cn)

### Abstract

To investigate the sources and evolution of haze pollution in different seasons, long-term (from 15 August to 4 December 2015) variations of chemical composition of $PM_1$ were characterized in Beijing, China. Positive matrix factorization (PMF) analysis with multi-linear engine (ME-2) resolved three primary and two secondary OA sources, including hydrocarbon-like OA (HOA), cooking OA (COA), coal combustion OA (CCOA), local secondary OA (LSOA) and regional SOA (RSOA). Distinctly different correlations between RSOA and sulfate were found in our study, with tight correlation ($R^2 = 0.71$) in late summer, decreased correlation ($R^2 = 0.62$) in autumn and almost no correlation ($R^2 = 0.02$) in early winter. This difference implies that sulfate was mainly transported at a large regional scale in late summer, while local and/or nearby sulfate formation may be more important in winter. Secondary aerosol species including SIA (sulfate, nitrate and



ammonium) and SOA (LSOA and RSOA) dominated $PM_1$ during all three seasons. In particular, SOA contributed 46% to total $PM_1$ (with 31% as RSOA) in late summer, whereas SIA contributed 41% and 45% to total $PM_1$ in autumn and early winter, respectively. Enhanced contributions of secondary species (66-76% of $PM_1$) were also observed in pollution episodes during all three seasons, further emphasizing the importance of secondary formation processes in haze pollution in Beijing. Combining chemical composition and meteorological data, our analyses suggest that photochemical oxidation dominated SOA formation during all three seasons, while for sulfate formation, gas-phase photochemical oxidation was the major pathway in late summer and heterogeneous processes were likely more important in autumn and early winter.

## 1. Introduction

Atmospheric particulate matter (PM) has broad impacts on the environment, including air quality (Sun et al., 2010; Sun et al., 2013; Huang et al., 2014), regional and global climate (Kaufman et al., 2002; IPCC, 2007), and human health (Pope et al., 2002; Lelieveld et al., 2015). Over the past decades, PM pollution in China has become one of the most serious environmental problem (Li et al., 2017). Beijing, the capital of China, has been suffering from severe haze events, with annual concentrations of $PM_{2.5}$ frequently exceeding the Chinese National Ambient Air Quality Standard (35 μg m$^{-3}$ as an annual average) (He et al., 2001; Streets et al., 2007; Huang et al., 2014; Wang et al., 2015). Effective mitigation of PM pollution requires a better understanding of the emission sources and atmospheric evolution processes (Cao et al., 2012; Huang et al., 2014; Guo et al., 2014; Sun et al., 2014).

The Aerodyne Aerosol mass spectrometers (AMS) have been widely used to obtain real-time measurements of the chemical composition of the non-refractory PM (NR-PM), including organic aerosol (OA), sulfate, nitrate, ammonium and chloride. Real-time techniques such as AMS overcome some limitations of offline techniques, for instance, measurement artifacts or limited time resolution (DeCarlo et al., 2006; Canagaratna et al., 2007; Ng et al., 2011). The Aerosol Chemical Speciation Monitor (ACSM), which is a simplified version of AMS, was designed for long-term measurements of NR-$PM_1$. In Beijing, a number of online and offline studies have been conducted in recent years to investigate the chemical composition, emission sources and formation mechanisms of PM (Chan and Yao, 2008; Zhao et al., 2013; Huang et al., 2014; Tian et al., 2014; Ho et al., 2015; Wang et al., 2015; Xu et al., 2015; Yang et al., 2015; Elser et al., 2016). It has been found that OA is the most dominant contributor to fine PM and that secondary aerosol plays an important role in haze formation (Huang et al., 2014; Elser et al., 2016).

Atmospheric receptor models, e.g., positive matrix factorization (PMF, Paatero and Tapper, 1994), have been successfully used to perform OA source apportionment based



on the OA mass spectral data (Lanz et al., 2007; Ulbrich et al., 2009; Sun et al., 2012, 2013; Elser et al., 2016; Wang et al., 2017). Primary OA (POA) sources such as hydrocarbon-like OA (HOA), cooking OA (COA) and biomass burning OA (BBOA) or coal combustion OA (CCOA) have been identified, while secondary OA (SOA) factors could be resolved either

based on oxidation state (i.e., less-oxidized oxygenated OA (LO-OOA) and more-oxidized oxygenated OA (MO-OOA)) or based on volatility (i.e., semi-volatility oxygenated OA (SV-OOA) and low-volatility oxygenated OA (LV-OOA)) (Huang et al., 2012; Crippa et al., 2013; Hu et al., 2013; Wang et al., 2017). PMF analyses have been used in a number of studies in Beijing (Huang et al., 2010; Sun et al., 2013, 2014, 2016, 2018; Huang et al., 2014; Elser et

al., 2016; Hu et al., 2016).

Despite a large number of studies aforementioned, the major sources and mechanisms responsible for the PM pollution during haze events are not well constrained, mainly due to complex interplay among local emission, regional transport, secondary reaction, as well as meteorological influence (Ma et al., 2010; Tao et al., 2012; Sun et al.,

2014; Zhang et al., 2017). For example, Hu et al. (2016) reported a stable ~80% contribution of secondary species to $PM_1$ in summertime Beijing, while $PM_1$ mass concentration in winter changed dramatically due to different meteorological conditions and enhanced primary emissions. However, Huang et al. (2014) and Elser et al. (2016) found that secondary aerosol formation also plays a crucial role in wintertime haze events

in Beijing. The formation mechanisms of secondary aerosol during haze events are not well constrained. Besides photochemical reactions, aqueous-phase reactions have been suggested to contribute to SOA formation. For example, PMF studies show that an aqueous OOA factor contributed 12% of total OA in wintertime Beijing and that the oxidation degree of OA increased at high RH levels (> 50%) (Sun et al., 2016). In combination with

the back-trajectory analysis, it is found that high $PM_1$ concentrations in Beijing were associated with air masses from the south and southwest and characterized by high fractions of MO-OOA and secondary inorganic aerosol, whereas direct emissions from local sources were the main contributor during clean events (Sun et al., 2015). These results show the inhomogeneity in the contribution to PM pollution depending on

different sampling locations and seasons, highlighting the need for more studies on chemical composition, sources and atmospheric evolution of PM.

In this study, we discuss the seasonal characteristics of chemical nature, sources, and atmospheric evolution of $PM_1$ in urban Beijing. Specifically, the formation mechanisms of secondary species and the impacts of meteorological conditions on the haze pollution are

elucidated.

## 2. Experimental

### 2.1 Measurement site





Measurements were conducted at an urban site in the National Center for Nanoscience (39.99°N, 116.32°E) in Beijing, which is close to the fourth ring of Beijing and surrounded by residential, commercial and traffic areas. All instruments were deployed on the roof of a five-story building (∼20 m above the ground) and the measurements were performed from 15 August to 4 December, 2015.

## 2.2 Instrumentation

NR-PM$_1$ species including organics, sulfate, nitrate, ammonium and chloride were continuously measured by an Aerodyne quadrupole ACSM (Q-ACSM) with a time resolution of ∼30 min. Detailed descriptions of ACSM operation can be found elsewhere (Ng et al., 2011a; Wang et al., 2017). Briefly, the ambient aerosol was sampled at a flowrate of ∼3 L min$^{-1}$ through a 3/8-inch stainless steel tube and an URG cyclone (Model: URG-2000-30ED) with a size cut of 2.5 μm in front of the sampling inlet was used to remove coarse particles. A Nafion dryer (MD-110-48S; Perma Pure, Inc., Lakewood, NJ, USA) was applied to dry aerosol particles before entering the ACSM and the submicron aerosol was subsampled into the ACSM with a flow rate of 85 cc min$^{-1}$ fixed by a 100 μm diameter critical aperture. The submicron particles were focused into a narrow beam by an aerodynamic lens and impacted a hot vaporizer (∼600 ℃). The resulting vapor was ionized with electron impact and chemically characterized with a quadrupole mass spectrometer. Mono-dispersed 300 nm ammonium nitrate particles, generated by an atomizer (Model 9302, TSI Inc., Shoreview, MN, USA) and selected by a differential mobility analyzer (DMA, TSI model 3080), were used to determine the response factor (RF) and calibrate the ionization efficiency (IE) (Ng et al., 2011a).

An Aethalometer (Model AE-33, Magee Scientific) was used for the determination of BC concentration with a time resolution of 1 min. SO$_2$ was measured by an Ecotech EC 9850 sulfur dioxide analyzer, CO by a Thermo Scientific Model 48i carbon monoxide analyzer, NOx by a Thermo Scientific Model 42i NO-NO$_2$-NOx analyzer and O$_3$ by a Thermo Scientific Model 49i ozone analyzer. Meteorological parameters, including wind speed, wind direction, relative humidity (RH), and temperature were measured by an automatic weather station (MAWS201, Vaisala, Vantaa, Finland) and a wind sensor (Vaisala Model QMW101-M2).

## 2.3 Data analysis

### 2.3.1. ACSM data analysis

The standard ACSM data analysis software in Igor Pro (WaveMetrics, Inc., Lake Oswego, Oregon USA) was used to analyze the ACSM dataset. IE was determined by comparing the response factors of ACSM to the mass calculated with the known particle



size and the number concentration from CPC. Standard relative ionization efficiencies (RIEs) were used for organics, nitrate and chloride (i.e., 1.4 for organics, 1.1 for nitrate and 1.3 for chloride) and RIEs for ammonium (6.4) and sulfate (1.2) were estimated from the IE calibrations using $NH_4NO_3$ and $NH_4SO_4$. The collection efficiency (CE) was introduced

to correct for the particle loss due to particle bounce, which is influenced by aerosol acidity, composition and the aerosol water content. As aerosol was dried before entering the ACSM, and particles are overall neutralized, the influences of particle phase water and acidity are expected to be negligible. Therefore, CE was determined as $CE_{dry}$ = max (0.45, 0.0833 + 0.9167 × ANMF), where ANMF represents the mass fraction of ammonium

nitrate in $NR-PM_1$ (Middlebrook et al., 2012).

### 2.3.2 Source apportionment

PMF was used to perform the source apportionment on the organic spectral data as implemented by the multilinear engine (ME-2; Paatero, 1997) via the interface SoFi (Source Finder) coded in Igor Wavemetrics (Canonaco et al., 2013). First, a range of

solutions with two to eight factors from unconstrained runs were examined. The POA factors mixed seriously with the SOA factors in the 3-factor solution, and there was no new interpretable factor when increasing the factor numbers above four in the PMF analysis. Therefore, the four-factor solution (HOA + CCOA, COA, OOA1 and OOA2) was adopted (Fig. S1). In the four-factor solution, the COA factor was well-defined through the

much higher contribution of $m/z$ 55 than $m/z$ 57 in its profile and the symbolic diurnal cycle of three peaks corresponding to the time of breakfast, lunch and dinner, supporting the assignment of the COA factor. Although the COA profile was well-defined, HOA and CCOA were totally mixed in the four-factor PMF solution, and the mixed factor had hydrocarbon-like fragments of $C_nH_{2n-1}$ and $C_nH_{2n+1}$ as in HOA but substantial amounts of

PAH-related ions as in CCOA. This mixed HOA + CCOA factor could not be further separated when increasing the number of factors, likely due to low mass resolution in ACSM data and limited capacity of PMF in separating similar factors. The mixture of HOA and CCOA factors was also observed in Sun et al. (2018), suggesting the difficulty in separating HOA and CCOA with PMF for the ACSM dataset. Compared to PMF, the ME-2

approach can direct the apportionment towards an environmentally-meaningful solution by introducing *a priori* information (profiles) for certain factors (Canonaco et al., 2013; Crippa et al., 2014; Frohlich et al., 2015). The ME-2 runs of five-factor were performed to separate HOA from CCOA and further optimize the apportionment solutions. We first constrained the HOA using HOA profile from Ng et al. (2011b), which is the average over

15 sites all over the world (including China, Japan, Europe and the United States). Previous studies have suggested that the HOA spectra from Europe and China are similar (Ng et al., 2011b; Elser et al., 2016) despite the different vehicle fuel patterns in China and Europe. When HOA was constrained, a new CCOA factor could be resolved. However, this CCOA





factor was seriously mixed with OOA as indicated by a relatively higher intensity at *m/z* 44 in the CCOA profile (Fig. S2). We thus further constrained the CCOA profile to decrease the influence of OOA on the CCOA factor. A CCOA profile from our previous study (Wang et al, 2017) was used to constrain CCOA. To minimize the effect from non-local input profiles (for both HOA and CCOA), the *a* value approach was used to adjust the input profiles to a certain extent. In addition, we also constrained COA profile from the 4-factor PMF solution with an *a* value of 0, which is a well-defined local profile as discussed above.

We tested *a* values for HOA and CCOA profiles between 0 and 1 with an interval of 0.1 and obtained 121 possible results, among which 6 solutions were reasonable based on the verification of the rationality of unconstrained factors, distinct mass spectra and time series, interpretable diurnal cycles and good correlations with external tracers for all factors. The final profiles and time series of individual factor were averaged from these 6 solutions and the standard deviations of intensities at each *m/z* was shown as error bars.

## 3. Results and discussion

### 3.1 Overview of mass concentration and chemical composition

Fig. 1 shows the time series of meteorological parameters, trace gases and $PM_1$ composition during the entire measurement period. The relatively clean events and polluted episodes occurred alternatively during the entire campaign. As shown in Fig. 1, the variations of $PM_1$ species are strongly associated with meteorological conditions. For example, clean periods were generally associated with northerly and northwesterly winds with high wind speeds. However, serious pollution episodes were related to southerly winds with low wind speeds ($< 1$ m s$^{-1}$), indicating the important role of stagnant meteorological conditions in haze pollution (Takegawa et al., 2009; Huang et al., 2010; Sun et al., 2014). The mass concentration of $PM_1$ varied from 0.4 µg m$^{-3}$ to 260.7 µg m$^{-3}$. Considering that the long-term measurements in our study have different meteorological conditions, we separated the entire study into three periods as late summer (15 August to 10 September), autumn (11 September to 10 November) and early winter (11 November to 4 December) in order to discuss the seasonal variations of $PM_1$ mass concentration and chemical composition.

The average mass concentration of $PM_1$ was 21.6 µg m$^{-3}$ in late summer (Fig. S3), which was much lower than that measured in July-August 2011 (50.0 µg m$^{-3}$, Sun et al., 2012) and in August-September 2011 (84.0 µg m$^{-3}$, Hu et al., 2016) (see Table 1). This lower $PM_1$ concentration was likely associated with the 2015 China Victory Day parade control from 23 August to 3 September, which significantly improved air quality in Beijing (Zhao et al., 2017). OA constituted a major fraction of $PM_1$ mass (64%), followed by sulfate (14%), BC (8%), ammonium (7%), nitrate (6%) and chloride (1%). During autumn, the mean concentration of $PM_1$ increased to 43.3 µg m$^{-3}$, which was two times higher than that





in late summer. OA contributed a mass fraction of 49%, followed by nitrate, sulfate, ammonium, BC and chloride with the mass fractions of 22%, 11%, 8%, 8% and 2%, respectively. Compared to late summer, the mass fraction of OA decreased to 49% (but the OA mass increased from 13.8 to 21.2 µg m$^{-3}$) and the mass fraction of inorganic species

increased correspondingly. The increase of inorganics was particularly noticeable for nitrate, which increased from 6% to 22% (or from 1.3 to 9.5 µg m$^{-3}$). The mean concentration of PM$_1$ was 64.3 µg m$^{-3}$ in early winter, further higher than those in late summer and autumn. This PM$_1$ average concentration in wintertime Beijing is similar with other studies such as Hu et al. 2016 (60.0 µg m$^{-3}$), Sun et al., 2013 (66.8 µg m$^{-3}$) and

Sun et al., 2016 (64.0 µg m$^{-3}$). OA accounted for 46% of PM$_1$ mass in early winter, followed by 20% of nitrate, 15% of sulfate, 10% of ammonium, 6% of BC and 3% of chloride (Fig S3).

As shown in Fig. 1f and Fig. S3, OA dominated PM$_1$ mass in late summer and autumn, whereas inorganic species played a more important role in early winter. It should also be

noted that nitrate had a more important contribution than sulfate to PM$_1$ during autumn and early winter, with nitrate/sulfate mass ratios of 2.0 and 1.3 in autumn and early winter, respectively. This phenomenon is likely due to the efficient emission reduction of SO$_2$ and the continuous increase of NO$_x$ because of dramatic growth of the vehicle fleets and large emissions from industries (Xu et al., 2015). Therefore, nitrate is expected to play

a more important role in PM pollution in the near future and controlling NO$_x$ emission would greatly help mitigating air pollution in Beijing.

The diurnal cycles of PM$_1$ species during different seasons are shown in Fig. S4. OA was characterized by three peaks occurring in the morning (06:00-09:00), at noon (12:00-14:00) and in the evening (19:00-22:00) during all three seasons. Such diurnal

patterns were partially influenced by the emission behavior of pollution sources, i.e., traffic, cooking and/or coal burning emissions (Huang et al., 2012; Sun et al., 2012; Crippa et al., 2013). Due to lower temperature in early winter, the planetary boundary layer (PBL) height was relatively flat compared to that in autumn and late summer, thus the noon peak of OA was more evident in early winter. The morning peak of OA was even more

pronounced than the noon peak in late summer. Such a diurnal cycle was likely related to the efficient photochemical oxidation in the morning and efficient dilution effect resulted from PBL height increase due to high temperature at noon.

The diurnal cycle of nitrate varied significantly during different seasons due to the seasonal difference in photochemical production and gas-particle partitioning (Sun et al.,

2015). Compared to nitrate, sulfate showed a relatively flat diurnal cycle in all seasons. A clear increase of sulfate in the afternoon was observed during late summer and autumn due to enhanced photochemical processes (Takegawa et al., 2009). In the winter, however, sulfate showed a decreasing trend in the afternoon, suggesting low photochemical production as discussed below. Chloride presented a morning peak and then rapidly





decreased to a low concentration level at ~18:00 during late summer, while in both autumn and winter, chloride displayed a diurnal cycle with higher concentrations at nighttime which may be related to the local emission from coal combustion. BC also showed the similar diurnal cycle with higher concentrations at nighttime and lower
concentrations in daytime during all three seasons.

### 3.2 Primary OA factors

Three POA factors were resolved in this study, including HOA, COA and CCOA. As shown in Fig. 2a, HOA mass spectrum is characterized by prominent hydrocarbon ion series of $C_nH_{2n-1}$ and $C_nH_{2n+1}$, particularly $m/z$ 27, 29, 41, 43, 55, 57, 67, 71. The HOA
spectrum is similar to previously reported HOA spectra at various urban sites (He et al., 2011; Ng et al., 2011; Sun et al., 2012). The time series of HOA is also correlated well with that of BC, which is an external tracer of incomplete combustion ($R^2$ = 0.56). The mass fractions of HOA (10-13%) and diurnal cycles in different seasons are rather consistent. There are two peaks from rush hours, i.e., 7:00-9:00 in the morning and around 20:00 in
the evening. The nighttime concentrations are generally high (Fig. S4), likely due to increased diesel fleets which are allowed in urban Beijing only at night.

The COA profile is characterized by prominent ion peaks at $m/z$ 55 and $m/z$ 57 (Fig. 2b), and a higher ratio of intensity at $m/z$ 55 over that at $m/z$ 57 (= 2.3) compared to the other two primary OA components (~1), which have been shown to be robust markers
for COA (He et al., 2010; Mohr et al., 2012; Crippa et al., 2013; Elser et al., 2016). This COA mass spectrum is highly correlated with other COA profile reported in previous studies (Crippa et al., 2013; Elser et al., 2016; Wang et al., 2017) and the time series correlated well with that of $m/z$ 55 with $R^2$ = 0.81. The COA diurnal cycle showed two obvious peaks at lunch (12:00) and dinner (20:00) time and a smaller peak at breakfast time (7:00) (Fig.
S4). Similar diurnal behaviors of COA have been observed in Beijing and other urban sites (Allan et al., 2010; Sun et al., 2010, 2013). COA had a lower mass fraction of 11% during late summer compared to autumn (20%) and early winter (16%).

The mass spectrum of CCOA is dominated by unsaturated hydrocarbons, particularly PAH-related ion peaks (e.g., 77, 91, and 115) (Dall'Osto et al., 2013; Hu et al., 2013). It
shows a similar spectral pattern with the ambient CCOA mass spectra in Beijing and Xi'an (Elser et al., 2016). The presence of CCOA can be further validated by the good correlation with external combustion tracer chloride ($R^2$ = 0.77) (Zhang et al., 2012). The time series of CCOA shows that the mass concentration of CCOA was much lower in August and September but increased dramatically after November, indicating the large emissions
from residential coal combustion for domestic heating. Also, the nighttime CCOA concentrations were much higher than the daytime concentrations, further confirming the enhanced coal combustion emissions from domestic heating in wintertime nights.





Specifically, on average, the mass fraction of CCOA increased from 5% (0.7 µg m⁻³) in late summer to 9% (2.0 µg m⁻³) in autumn and then to 26% (7.7 µg m⁻³) in early winter (Fig. S3).

### 3.3 Secondary OA factors and sulfate sources: regional transport v.s. local formation

Two oxygenated OA factors were identified in our study, i.e., local SOA (LSOA) and regional SOA (RSOA) (Fig. 2). These two SOA factors show similar mass spectra with high ratios of intensity at $m/z$ 44 over that at $m/z$ 43 ($f_{44/43}$), but their time series differ greatly. The time series of LSOA is highly correlated with that of nitrate in the entire period with $R^2$ = 0.83, whereas there is no good correlation ($R^2$ = 0.11) between time series of RSOA and nitrate, indicating that they are two factors related to different sources/atmospheric processes. The $f_{44/43}$ of RSOA (4.8) is higher than that of LSOA (2.9), suggesting that RSOA from regional transport is more oxygenated (more aged) than locally formed SOA (Sun et al., 2014, 2015). The average mass concentration of LSOA increased from 3.2 µg m⁻³ in late summer to 9.2 µg m⁻³ in autumn and to 12.1 µg m⁻³ in early winter with an increase of mass fraction from 23% in late summer to 43% in autumn and 41% in early winter. On the contrary, the average mass concentration of RSOA decreased from 6.6 µg m⁻³ in late summer to 3.8 µg m⁻³ in autumn and to 1.8 µg m⁻³ in early winter, with the dramatic decrease of mass fraction from 48% in late summer to 18% in autumn and to 6% in early winter (Fig. S3). These seasonal variations of LSOA and RSOA indicate that RSOA related to regional transport was more important during late summer, while locally formed LSOA played a dominant role in autumn and early winter.

In our study, different correlations between sulfate and RSOA or LSOA were found during different seasons. As shown in Fig. 3b, time series of RSOA correlated well with that of sulfate during late summer with $R^2$ = 0.71. This correlation coefficient decreased to 0.62 during autumn and there was almost no correlation between RSOA and sulfate ($R^2$ = 0.02) in early winter. On the contrary, the correlations between LSOA and sulfate displayed the opposite variation with the correlation coefficient ($R^2$) increased from 0.40 in late summer to 0.66 in autumn and 0.86 in early winter (Fig. 3a). As RSOA is related to regional source and LSOA indicates local source, different correlations between sulfate and RSOA or LSOA (i.e., better correlation with RSOA in late summer, similar correlation with RSOA and LSOA in autumn and tight correlation with LSOA in early winter) may indicate that sulfate have different source regions during different seasons due to the change of meteorological conditions. In order to further analyze sources of sulfate in our study period, the bivariate polar plots of sulfate during different seasons are displayed in Fig. 3c. During late summer, the high mass concentration of sulfate mainly located in the south and southwest regions from the sampling site thus the correlation coefficient



between sulfate and RSOA ($R^2$ = 0.71) was higher than that between sulfate and LSOA ($R^2$ = 0.40). Regional transport was the major source to sulfate formation in late summer. However, high sulfate located both at the sampling site and in the south regions from sampling site in autumn, which suggests that both local formation and regional transport

contributed to the sulfate concentration, thus there was a similar correlation between sulfate and RSOA and between sulfate and LSOA with $R^2$ values of 0.62 and 0.66, respectively. When it comes to the early winter, high mass concentrations of sulfate mainly located in the sampling site coming from local formation and there was almost no contribution from regional transport. Consistently, sulfate was tightly correlated with

LSOA ($R^2$ = 0.86) during early winter and there was no correlation between sulfate and RSOA ($R^2$ = 0.02). These results indicate that sulfate transported at a large regional scale was more important during late summer, while local formation was the major source of sulfate in early winter due to residential heating.

### 3.4 Contribution of secondary species to PM pollution

The average $PM_1$ concentration increased from late summer (21.6 µg m$^{-3}$) to early winter (64.3 µg m$^{-3}$) (Fig. S3) and the chemical composition showed seasonal difference. The mass concentrations of secondary species increased from 15.7 µg m$^{-3}$ in late summer to 30.8 µg m$^{-3}$ in autumn and to 42.8 µg m$^{-3}$ in early winter, but the mass fraction in $PM_1$ decreased from 72% in late summer to 66% in early winter. In particular, SOA had a

dominant contribution in late summer (9.8 µg m$^{-3}$, 46% of $PM_1$), while SIA played a key role during autumn (17.8 µg m$^{-3}$, 41% of $PM_1$) and early winter (28.9 µg m$^{-3}$, 45% of $PM_1$) (Fig. S3). The high SOA fraction in summer is likely associated with active photochemical oxidation, while the increased SIA fraction in autumn and early winter is likely due to enhanced gas-particle partitioning of nitrate and aqueous-phase formation of sulfate.

Fig. 4 shows the $PM_1$ composition and OA sources in clean days (daily average $PM_1$ < 20 µg m$^{-3}$), medium pollution days (M-pollution, 40 µg m$^{-3}$ < daily average $PM_1$ < 80 µg m$^{-3}$) and high pollution days (H-pollution, daily average $PM_1$ > 80 µg m$^{-3}$) during different seasons. The mass concentrations of $PM_1$ species and OA factors, gaseous pollutants and meteorological parameters during different periods are summarized in Table S1. The

average concentration of $PM_1$ was 46.9 µg m$^{-3}$ during M-pollution days, about 3 times higher than that during clean days (15.6 µg m$^{-3}$) in late summer. In autumn and early winter, the average $PM_1$ concentrations during H-pollution days (110.5 µg m$^{-3}$ and 109.7 µg m$^{-3}$, respectively) were two times higher than those in M-pollution days (54.2 µg m$^{-3}$ and 43.5 µg m$^{-3}$, respectively) and ten times higher than those in clean days (9.3 µg m$^{-3}$

and 8.1 µg m$^{-3}$, respectively). As shown in Fig. 4, the mass fraction of secondary aerosol species (SIA and SOA) increased from clean days (52-70%) to M-pollution days (67-76%) and H-pollution days (66-74%) during all three seasons, emphasizing the significant





enhancements of secondary aerosol formation in haze pollution events (Huang et al., 2014; Jiang et al., 2015; Zheng et al., 2015). In late summer, the mass concentration of LSOA increased from 2.2 µg m$^{-3}$ (21% of OA) during clean days to 6.7 µg m$^{-3}$ (24% of OA) during M-pollution days and the mass concentration of RSOA increased from 5.0 µg m$^{-3}$ (48% of

OA) during clean days to 13.8 µg m$^{-3}$ (49% of OA) during M-pollution days, suggesting that regional transport played a more important role than local formation in both clean and haze pollution events during late summer. The mass concentration of LSOA increased from 1.5 µg m$^{-3}$ in clean days to 10.2 µg m$^{-3}$ in M-pollution days and to 25.4 µg m$^{-3}$ in H-pollution days during autumn and increased from 1.5 µg m$^{-3}$ in clean days to 7.5 µg m$^{-3}$ in

M-pollution days and to 20.7 µg m$^{-3}$ in H-pollution days during early winter. In comparison, the mass concentration of RSOA increased from 1.5 µg m$^{-3}$ and 0.6 µg m$^{-3}$ in clean days to 5.9 µg m$^{-3}$ and 2.0 µg m$^{-3}$ in M-pollution days and to 6.6 µg m$^{-3}$ and 2.5 µg m$^{-3}$ in H-pollution days during autumn and early winter, respectively. The increase rates of LSOA were much higher than that of RSOA, thus the mass fraction of LSOA increased

dramatically from clean days to M-pollution and H-pollution days in autumn and early winter (i.e., 26% to 40% and 50% during autumn and 33% to 37% and 42% during early winter), whereas the mass fraction of RSOA decreased from clean days to M-pollution and H-pollution days (i.e., 25% to 23% and 13% during autumn and 14% to 10% and 5% during early winter). These observations suggest that locally formed SOA had more

important contributions than regional sources in haze pollution during autumn and early winter, implying different contribution of secondary aerosol in different seasons.

### 3.5 Episodic analysis and meteorological effects

The clean and pollution episodes occurred in "saw-tooth cycles", in which meteorological conditions, regional transport, local emissions and secondary formation

intertwine and play different roles in the evolution of PM pollution. To get a better insight into aerosol sources and atmospheric processes, seven clean episodes, seven M-pollution episodes and five H-pollution episodes were selected for further analysis. As shown in Fig. 5, the pollution episodes were generally associated with higher RH and lower wind speeds (< 1 m s$^{-1}$) than that in clean episodes in autumn and early winter, with RH usually higher

than 60% in pollution episodes (both M-pollution and H-pollution) and lower than 45% in clean episodes. Specifically, an M-pollution (M1, 47.6 µg m$^{-3}$) episode in late summer had similar RH and wind speed with the adjacent clean period (C1, 14.1 µg m$^{-3}$). However, the contribution of organic species decreased from 68% in C1 to 61% in M1 but the mass fraction of secondary inorganic species (particularly sulfate) increased from 23% in C1 to

33% in M1. This phenomenon may result from enhanced photochemical formation of secondary species in M1 due to higher oxidation capacity as M1 had higher O$_3$ concentration (54.1ppb) than C1(31.0ppb). In autumn, the mass concentrations of





organics increased from 4.8-6.3 µg m$^{-3}$ during C2-C5 to 21.2-27.8 µg m$^{-3}$ during M2-M6 while the contributions decreased from 56-71% to 39-55%, and the corresponding contributions of secondary inorganic species increased from 17-29% during C2-C5 to 36-52% during M2-M6 with mass concentrations increased from 1.6-2.9 µg m$^{-3}$ to 16.7-33.1µg m$^{-3}$. The contributions of secondary organic species to OA also increased from 50-61% to 55-73% with mass concentrations increased from 2.7-3.6 µg m$^{-3}$ to 14.1-19.4 µg m$^{-3}$. This indicates a notable production and accumulation of secondary aerosol during pollution events. Compared to M-pollution episodes, there was no further increase of contribution of secondary inorganic species during H1-H3 (42-47%) although the mass concentrations increased to 45.3-56.6 µg m$^{-3}$ due to the systematic concentration growths of all species from M-pollution to H-pollution. Secondary organic species also had similar contributions to OA during H1-H3 (52-75%) with that during M2-M6 (55-73%) although the mass concentrations increased from 14.1-19.4 µg m$^{-3}$ to 25.6-38.5 µg m$^{-3}$. Further analysis shows that the RH during H1-H3 (71.7%-81.6%) is lower than that during M2-M6 (74.1%-91.8%), which indicates that the stronger aqueous-phase chemistry during M2-M6 may lead to the efficient formation of secondary species and the mass concentration growths of secondary species were faster than that of other species in PM$_1$ thus the mass fraction of secondary species in M2-M6 were higher or similar with that in H1-H3. A similar phenomenon was also found in early winter. The contributions of secondary species increased from clean episodes (C6 and C7) to pollution episodes (M7, H4 and H5) while the contributions of secondary species were similar in M7, H4 and H5 because of similar RH. These PM evolution characteristics observed here highlight the importance of meteorological conditions on driving particulate pollution (Li et al., 2017) and imply different formation mechanisms of PM pollution during different seasons.

### 3.6 Photochemical oxidation and aqueous-phase chemistry

To further elucidate the formation mechanisms of secondary aerosol, the sulfur oxidation ratio ($F_{SO_4}$) (Sun et al., 2006) was calculated according to Eq. (1):

$$F_{SO_4} = \frac{n[SO_4]}{n[SO_4]+n[SO_2]} \qquad (1)$$

where $n[SO_4]$ and $n[SO_2]$ are the molar concentrations of sulfate and SO$_2$, respectively. Fig. 6a-c plots F$_{SO4}$ verse RH, colored by O$_x$ (= O$_3$ + NO$_2$) concentration which is a tracer to indicate photochemical processing during late summer, autumn and early winter, respectively. F$_{SO4}$ presents an evident exponential relationship with RH in autumn and early winter, and the relationship in early winter is even more pronounced, suggesting that aqueous-phase formation of sulfate might play an important role during extreme haze events in autumn and early winter in Beijing (Sun et al., 2013; Elser et al., 2016). However, Fig. 6a shows that in late summer F$_{SO4}$ increased with RH at RH < 60% then





decreased with RH at RH > 60%, which is different from those in autumn and early winter. When taking $O_x$ into account, it is found that $O_x$ reached the peak concentration when RH was 50-60%, then decreased when RH was > 60%. The different characteristic of $F_{SO4}$ in late summer was likely due to the large influence from photochemical oxidation. The

relationship between $F_{SO4}$ and $O_x$ during different seasons are also shown in Fig. 6d-f. There is a liner relationship between $F_{SO4}$ and $O_x$ in the whole RH range during late summer with $R^2 = 0.40$. A liner relationship between $F_{SO4}$ and $O_x$ could still be observed in autumn at RH < 65% with $R^2 = 0.48$. However, there is no clear relationship between $F_{SO4}$ and $O_x$ concentration in early winter, although there is a weak increasing trend for a few

data points with $O_x$ concentration higher than 40 ppb. These results suggest that photochemical oxidation played an important role for sulfate formation in late summer, while aqueous-phase reactions were more responsible for the sulfate concentrations in autumn and early winter.

We further investigated the formation mechanisms of SOA during different seasons.

Fig. 7 shows the effects of RH and $O_x$ on the mass concentrations and mass fractions of LSOA and RSOA during different seasons. The variations of OA composition as functions of RH was different among three seasons. During late summer, both the mass concentrations of LSOA and RSOA first increased at RH < 60% and then decreased as RH increased, which is similar with that between $F_{SO4}$ and RH. The mass concentration of

LSOA increased from 3.9 µg m⁻³ to 5.6 µg m⁻³ when RH increased from 20% to 60% and then decreased to 1.7 µg m⁻³ when RH increased to 90%. Similarly, the mass concentration of RSOA increased from 6.4 µg m⁻³ to 11.9 µg m⁻³ and then decreased to 4.2 µg m⁻³. This suggest that the high RH condition did not promote the formation of SOA and the mass decrease is likely due to the concentration decrease of $O_x$ when RH > 60% (as shown in

Fig. 6). During autumn and early winter, mass concentrations of LSOA and RSOA increased gradually as RH increased and this increasing trend become flat when RH >60%. Moreover, the mass fraction of SOA did not show clear changes when RH increased. Variations of the mass concentrations and fractions of LSOA and RSOA as functions of $O_x$ during different seasons were also shown in Fig. 7. The mass concentrations of SOA

increased clearly with the increase of $O_x$ concentration during all three seasons and the mass fraction of SOA also increased from 64% to 76% during late summer and increased from 59% to 80% during autumn as $O_x$ increased from 30 ppb to 120 ppb. However, the increasing rates of LSOA and RSOA as functions of $O_x$ were substantially different among different seasons. In late summer, both LSOA and RSOA presented linear increases with

the increase of $O_x$. As a comparison, LSOA showed higher increase rates with $O_x$ than that of RSOA during autumn and early winter as LSOA played a dominant role in the haze formation during autumn and early winter. As shown in Fig. 7, SOA concentrations increased with $O_x$ in the whole range while decreased with RH at RH > 60% during late summer. The average increasing rates with $O_x$ was 4.4 µg m⁻³ per 10 ppb $O_x$ increase in



autumn and 5.6 μg m$^{-3}$ per 10 ppb O$_x$ increase in early winter for SOA (LSOA+RSOA) concentration. In comparison, the average increasing rates with RH was 1.7 μg m$^{-3}$ per 10% RH increase in autumn and 2.2 μg m$^{-3}$ per 10% RH increase in early winter. The increasing rates with O$_x$ are faster than that with RH during autumn and early winter. These results clearly indicate that photochemical processing played a dominant role in the formation of SOA during all three seasons although aqueous-phase chemistry may also have a contribution for the SOA formation during autumn and early winter.

## 4. Conclusion

In this study, an ACSM combined with an Aethalometer were applied for real-time measurements of PM$_1$ species (organics, sulfate, nitrate, ammonium, chloride and BC) from 15 August to 4 December, 2015 in Beijing. The average mass concentration of PM$_1$ varied from 21.6 μg m$^{-3}$ in late summer to 64.3 μg m$^{-3}$ in early winter, indicating that PM pollution was much serious in wintertime due to enhanced emissions, low temperatures and stagnant meteorological conditions. OA contributed the major fraction (46%-64%) to PM$_1$ mass during all three seasons, followed by nitrate (6%-22%) or sulfate (11%-15%). Regarding the OA factors, three primary OA (HOA, COA and CCOA) and two secondary OA (LSOA and RSOA) were resolved. Seasonal variations suggested that SOA dominated OA during late summer and autumn, whereas POA played a more important role in early winter due to the dramatically increased fraction of CCOA in heating season (from 5% in late summer to 26% in early winter). A higher RSOA fraction (48% of OA) in late summer and higher LSOA fractions in autumn (43% of OA) and early winter (41% of OA) and different correlations between RSOA and sulfate were found in our study, suggested that regional transport played a more important role in SOA and sulfate source in late summer, while local formation was important in winter due to heating.

Haze evolution and formation mechanisms of PM$_1$ were also discussed. Results suggested that secondary aerosol species including SIA (sulfate, nitrate and ammonium) and SOA (LSOA and RSOA) dominated PM$_1$ species during all three seasons with fractions of 72%, 71% and 66% during late summer, autumn and early winter, respectively. SOA had a dominant contribution to PM$_1$ in late summer, while SIA played a key role during autumn and early winter. Higher contributions of secondary species (SIA and SOA) further observed in pollution episodes emphasized the importance of the secondary formation processes in haze pollution in Beijing. We explored the formation mechanisms of secondary aerosol during different seasons and found that photochemical processing dominates SOA formation during all three seasons. In comparison, gas-phase photochemical oxidation was the major formation mechanism of sulfate in late summer, while aqueous-phase chemistry was likely playing an important role in autumn and early winter.



*Data availability.* Raw data used in this study are archived at the Institute of Earth Environment, Chinese Academy of Sciences, and are available on request by contacting the corresponding author.

5 *Supplement.* The Supplement related to this article is available online at

*Author contributions.* RJH and JC designed the study. JD, YG, YW and HZ performed the online measurements. Data analysis and source apportionment were done by JD, RJH and CL. JD and RJH wrote the manuscript. JD and RJH interpreted data and prepared display items. All authors commented on and discussed the manuscript.

10 *Competing interests.* The authors declare that they have no conflict of interest.

*Acknowledgements.* This work was supported by the National Natural Science Foundation of China (NSFC) under grant No. 91644219 and No. 41877408, and the National Key Research and Development Program of China (No. 2017YFC0212701).

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





**Table1.** Summary of $PM_1$ mass concentrations and composition as well as OA composition in Beijing during different seasons.

| Year | Season (Characteristic) | $PM_1$ ($\mu g\ m^{-3}$) | % of $PM_1$ | | | | | | % of OA | | Reference |
|---|---|---|---|---|---|---|---|---|---|---|---|
| | | | OA | $SO_4$ | $NO_3$ | $NH_4$ | Chl | BC | POA | SOA | |
| 2008 | Summer (Olympic Games) | 63.1 | 38 | 27 | 16 | 16 | 1 | 3 | 43 | 57 | Huang et al., 2010 |
| 2010 | Winter | 60.0 | 50 | 13 | 10 | 11 | 8 | 9 | 69 | 31 | Hu et al., |
| 2011 | Summer | 84.0 | 31 | 26 | 20 | 16 | 1 | 5 | 35 | 65 | 2016 |
| 2011 | Summer | 50.0 | 40 | 18 | 25 | 16 | 1 | - | 36 | 64 | Sun et al., 2012 |
| 2011 | Winter | 66.8 | 52 | 14 | 16 | 13 | 5 | - | 69 | 31 | Sun et al., 2013 |
| 2011 | Autumn | 53.3 | 50 | 12 | 21 | 13 | 3 | - | - | - | |
| 2011 | Winter | 58.7 | 51 | 13 | 17 | 14 | 5 | - | - | - | Sun et al., |
| 2012 | Spring | 52.3 | 41 | 14 | 25 | 17 | 3 | - | - | - | 2015 |
| 2012 | Summer | 61.6 | 40 | 17 | 25 | 17 | 1 | - | - | - | |
| 2012 | Winter (Non-Heating) | 56.0 | 48 | 12 | 18 | 9 | 4 | 9 | 45 | 55 | Wang et al., 2015 |
| 2012 | Winter (Heating) | 84.2 | 50 | 16 | 12 | 9 | 7 | 7 | 62 | 38 | |
| 2013 | Winter | 64.0 | 60 | 15 | 11 | 8 | 6 | - | 57 | 43 | Sun et al., 2016 |
| 2014 | Autumn (Before APEC) | 88.0 | 38 | 14 | 26 | 11 | 4 | 7 | 46 | 54 | Xu et al., 2015 |
| 2014 | Autumn (During APEC) | 41.6 | 52 | 9 | 19 | 9 | 5 | 6 | 66 | 34 | |
| 2015 | Autumn (Parade control) | 19.4 | 55 | 18 | 12 | 8 | 1 | 6 | 35 | 65 | Zhao et al.,2017 |
| 2015 | Autumn (Non-Parade Control) | 45.4 | 40 | 20 | 20 | 12 | 2 | 6 | 35 | 65 | |
| 2015 | Late Summer | 21.6 | 64 | 14 | 6 | 7 | 1 | 8 | 29 | 71 | |
| 2015 | Autumn | 43.3 | 49 | 11 | 22 | 8 | 2 | 8 | 39 | 61 | This Paper |
| 2015 | Early Winter | 64.3 | 46 | 15 | 20 | 10 | 3 | 6 | 53 | 47 | |



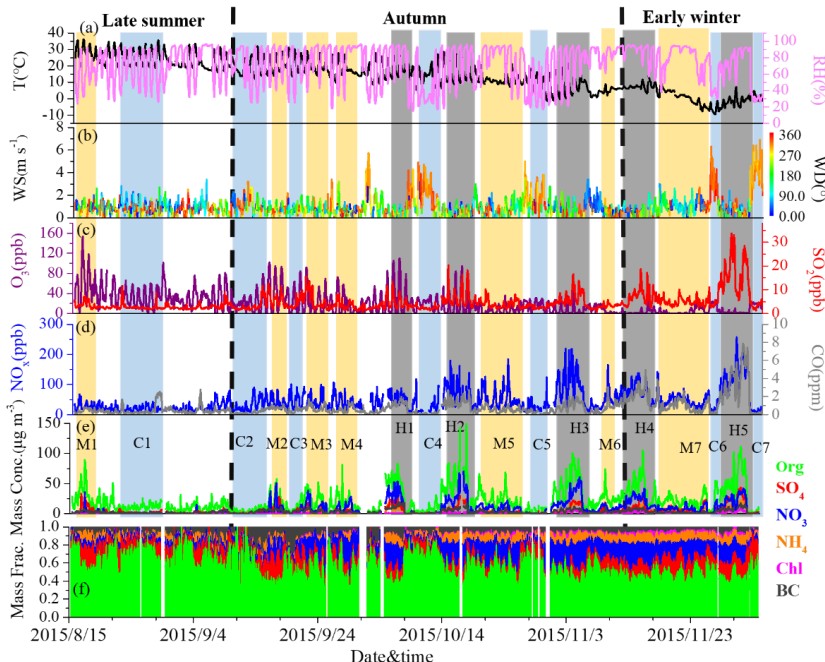

**Figure 1.** Time series of (a) temperature (T) and relative humidity (RH), (b) wind speed (WS) and wind direction (WD), (c) $O_3$ and $SO_2$, (d) CO and NOx, (e) $PM_1$ species, (f) mass fractions of $PM_1$ species during the entire study. 7 clean episodes (C1-C7), 7 moderate-pollution episodes (M1-M7) and 5 high-pollution episodes (H1-H5) are marked for further discussion.





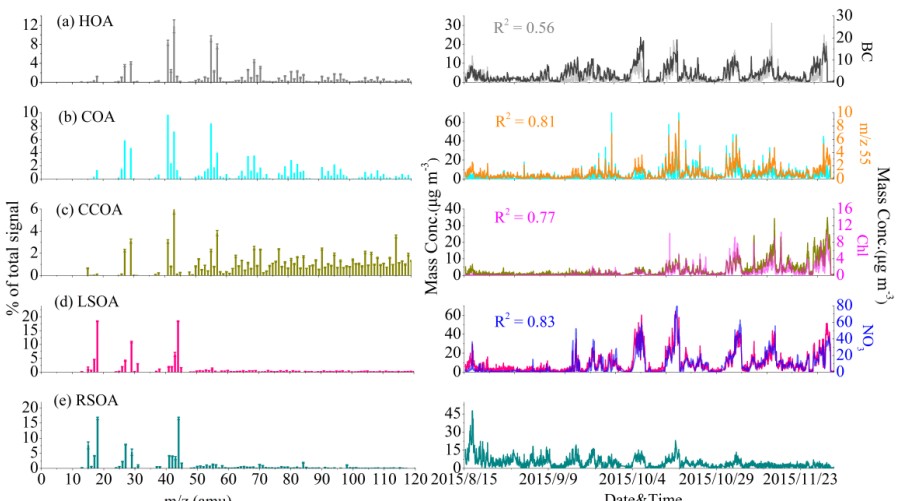

**Figure 2.** Mass spectrums (left) and time series (right) of five resolved OA factors. Error bars of mass spectrums represent the standard deviation of each *m/z* over all accepted solutions. The time series of BC, *m/z* 55, chloride and nitrate are shown for comparisons.



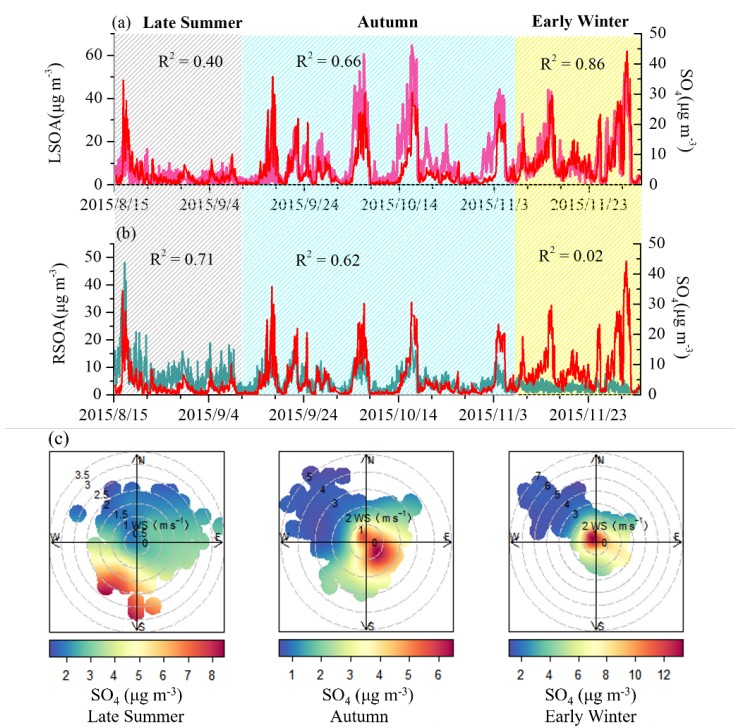

**Figure 3.** (a) correlation between time series of SO$_4$ and LSOA, (b) correlation between time series of SO$_4$ and RSOA, (c) bivariate polar plots of SO$_4$ during late summer (left), autumn (middle) and early winter (right) as functions of wind direction and wind speed (m s$^{-1}$).





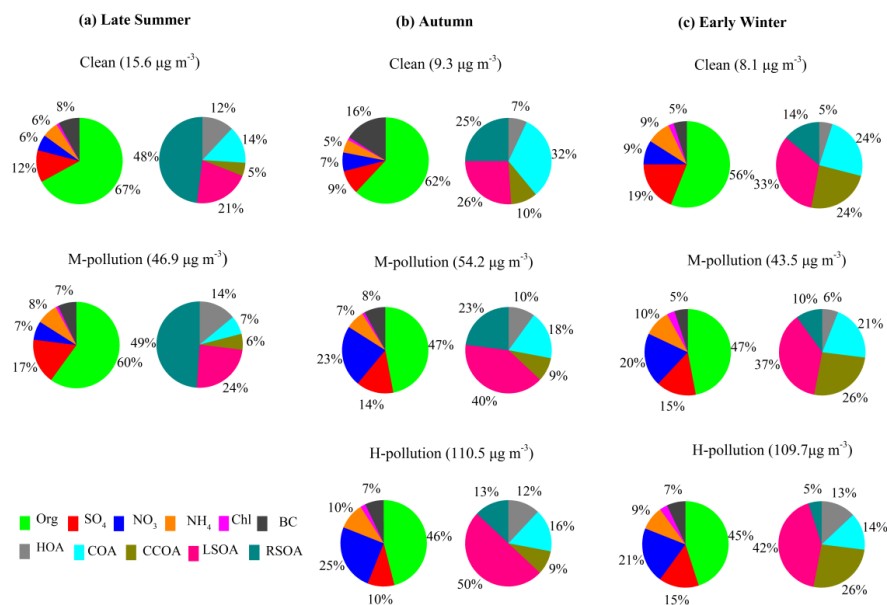

**Figure 4.** Relative contributions of PM$_1$ species and OA sources in clean days, M-pollution days and H-pollution days during late summer (a), autumn (b) and early winter (c).





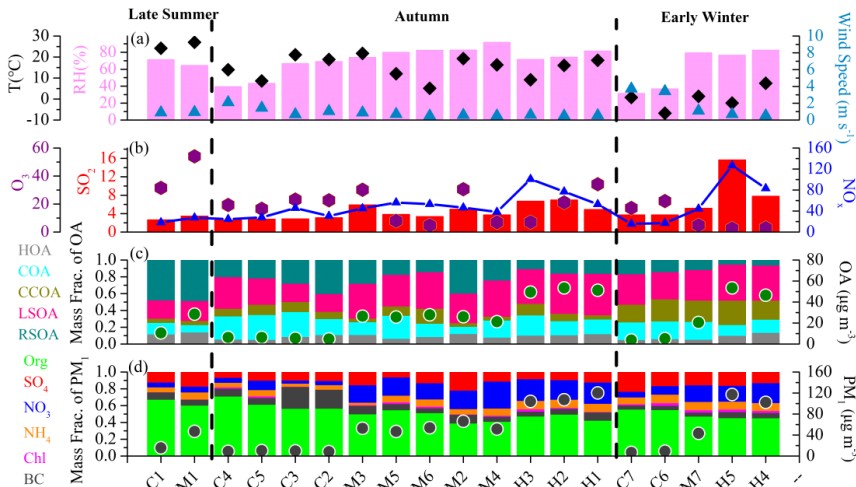

**Figure 5.** Summary of (a) meteorological parameters (RH, T, WS), (b) gaseous species (SO$_2$, NO$_x$, O$_3$), (c) OA factors and (d) PM$_1$ composition for episodes C1-C7, M1-M7 and H1-H5.





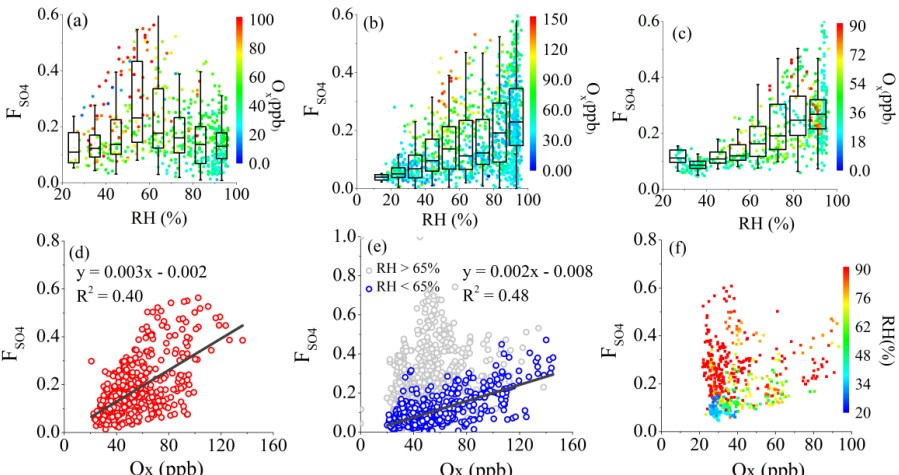

**Figure 6.** The relationship between sulfur oxidation ratio ($F_{SO4}$) and RH colored by $O_x$ concentration during late summer (a), autumn (b) and early winter (c) and the relationship between $F_{SO4}$ and $O_x$ during later summer (d), autumn(e) and early winter(f).





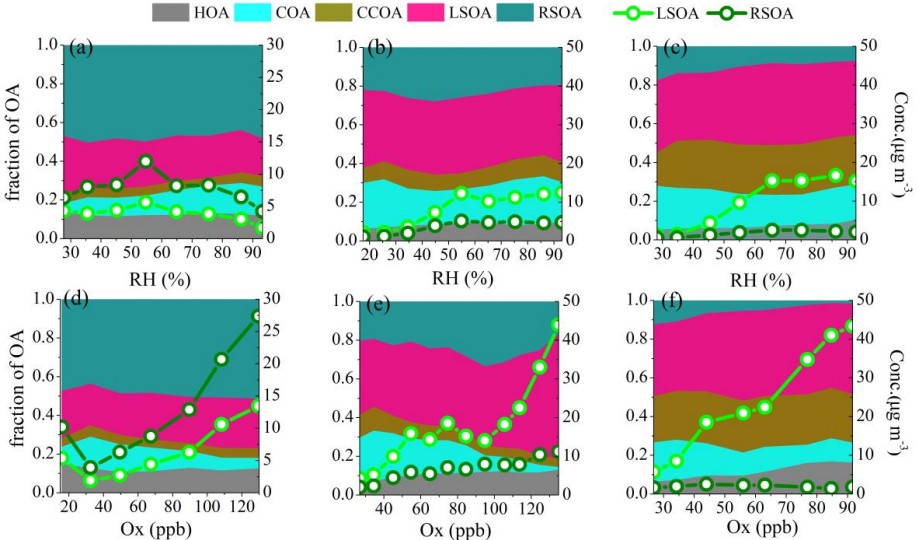

**Figure 7.** Variations of the mass fractions and mass concentrations of LSOA, RSOA as functions of RH or $O_x$ in (a, d) late summer, (b, e) autumn and (c, f) early winter. The data were binned according to the RH (10% increment) and $O_x$ concentration (20 ppb increment in late summer, 10 ppb increment in autumn and early winter).