# Peer review of "Distinctions in source regions and formation mechanisms of secondary aerosol in Beijing from summer to winter"

_Atmospheric Chemistry and Physics, 2019_

## Referee Comment (RC1) · Anonymous Referee #2 · 5 Apr 2019

In this study, the authors performed four-month measurement of the chemical composition of PM1 in Beijing, with an Aerosol Chemical Speciation Monitor (ACMS). PMF analysis is applied to study sources of organic aerosol. The authors perform routine analysis to study the regional vs. local sources of PM1, seasonal variation of PM1, and gas-phase vs. aqueous-phase formation pathways of organics and sulfate. The analysis procedure has been widely used in the literature. There have been numerous studies with ACSM or AMS to study the same topic in the same area (as summarized in Table 1). My overall impression is that this manuscript fails to establish enough novelty to distinguish itself from previous studies. Thus, I would not recommend this manuscript for publication in its current state.

[Figure]

The discussions on regional SOA factor (RSOA) and local SOA factor (LSOA) suffers major flaws in its logical flow. To start with, it is not justified why these two OA factors are classified as regional and local. In section 3.3, the authors simply state that two oxygenated OA factors were identified and they are local SOA and regional SOA, but no justification is provided. Moreover, the assumption that the one factor is regional and the other factor is local is applied throughout the manuscript to infer the sources of sulfate (P9 L25-35), OA (P11 L1-5), and PM1. However, this assumption is not justified yet! In P10 L1-15, bivariate polar plots of SO4 are used to discuss the sources of SO4. The results are reasonable. However, the sources of SO4 need to be established first and further used to infer the sources of LSOA and RSOA, instead of the other way around.

I also have concerns on the SO4 discussions. (1) It is important to note that particle liquid water content is a better proxy for aqueous phase reaction than RH. It has been well-established that the particle LWC not only depends on RH, but also on particle composition. (2) Figure 3a suggests that regional transport is the major source of SO4 in later summer. However, the good correlation between SO4 and Ox in figure 6 suggests SO4 may be locally formed. How to reconcile this difference?
* * *

---

## Referee Comment (RC2) · Anonymous Referee #1 · 8 Apr 2019

The manuscript presents online aerosol measurements performed at Beijing in three seasons and discusses about aerosol sources and formation products. It also evaluates the importance of primary vs secondary species and of photochemistry vs aqueous phase processes, in clean and polluted conditions in the three investigated seasons.

The manuscript is well written and the results appear robust. The topic can be considered adequate for ACP and of interest for the scientific community. I recommend publication after the following (major) comments have been addressed.

General comments

[Figure]

The weakest point of the data discussion is the characterization of the OOAs. The authors attribute one to regional processes (RSOA) and the other one to local processes (LSOA), but this attribution is not adequately supported in the manuscript. Considering that all the discussion is based on this attribution, the authors should be more convincing under this aspect. The authors present polar plots, showing the spatial distribution of sulfate sources, resulting in a credible distinction between local and regional sulfate. I invite them at least to present the same elaborations for RSOA e LSOA as well.

I do not see the utility of Par. 3.5 in the manuscript. First, it is not clear how the selected episodes have been identified. The authors should provide the criteria that lead to select these episodes instead of others. This would help the reader in understanding the discussion. Most importantly, the conclusions derived in this section appear largely speculative, as they are based on the comparison of one episode with a couple of others, which lacks of any statistical robustness. If the authors are interested in evaluating the effect of meteorology on the occurrence of pollution episodes, they should work with the whole dataset in a statistically robust way.

The analysis of the SOA production routes is very interesting, but RH is certainly not the best tracer for aqueous phase processes. I invite the authors to make use of the aerosol Liquid Water Content (LWC) instead, which is a much better tool for this purpose. It is not difficult to calculate the aerosol LWC, based on simple models, once RH, T and aerosol chemical composition are available.

Specific comments

P7.L13. "OA dominated PM1 mass in late summer and autumn, whereas inorganic species played a more important role in early winter". This sentence is not supported by the results: the difference in OA contribution between autumn and early winter is almost negligible (49 vs 46%). Fig. S3 shows clearly that OA is the dominant component in early winter as well.

P7.L27. It is not properly temperature that drives the boundary layer evolution. A lower

temperature is a consequence of the lower solar radiation reaching the surface, as it is a shallower boundary layer.

P8.L15-16. Also PBL dynamics may have an effect on this.

Figure 3a. Please change the colors of the plotted lines. It is hard to distinguish one line from the other.

[Figure]

---

## Author Comment (AC1) · 8 Jun 2019

The authors thank the editor and referees to review our manuscript and particularly for the valuable comments and suggestions that have significantly improved the manuscript. We provide below point-by-point responses to the referees' comments. We also have made most of the changes suggested by the referees in the revised manuscript.

Referee #1

The manuscript presents online aerosol measurements performed at Beijing in three seasons and discusses about aerosol sources and formation products. It also evaluates the importance of primary vs secondary species and of photochemistry vs aqueous phase processes, in clean and polluted conditions in the three investigated seasons.

The manuscript is well written and the results appear robust. The topic can be considered adequate for ACP and of interest for the scientific community. I recommend publication after the following (major) comments have been addressed.

General comments

1. The weakest point of the data discussion is the characterization of the OOAs. The authors attribute one to regional processes (RSOA) and the other one to local processes (LSOA), but this attribution is not adequately supported in the manuscript. Considering that all the discussion is based on this attribution, the authors should be more convincing under this aspect. The authors present polar plots, showing the spatial distribution of sulfate sources, resulting in a credible distinction between local and regional sulfate. I invite them at least to present the same elaborations for RSOA e LSOA as well.

Response: We thank the referee for this good point. Following the referee's suggestion, we have now included the spatial distribution of LSOA and RSOA. The polar plots below show clearly that LSOA is mainly located in the sampling site while RSOA is mainly from the south to the sampling site. Also, to be more logical, as suggested by referee #2, we have now reconstructed this part by discussing the source regions of sulfate before SOA.

[Figure]

In the revised manuscript in page 10, lines 22-24, we have now added the following discussion:

"sulfate...than locally formed SOA (Sun et al., 2014, 2015). The attribution of LSOA and RSOA is further supported by the bivariate polar plots (Fig. S5), which show clearly that LSOA is mainly located in the sampling site while RSOA is mainly from the south to the sampling site."

2.  I do not see the utility of Par. 3.5 in the manuscript. First, it is not clear how the selected episodes have been identified. The authors should provide the criteria that lead to select these episodes instead of others. This would help the reader in understanding the discussion. Most importantly, the conclusions derived in this section appear largely speculative, as they are based on the comparison of one episode with a couple of others, which lacks of any statistical robustness. If the authors are interested in evaluating the effect of meteorology on the occurrence of pollution episodes, they should work with the whole dataset in a statistically robust way.

Response: In Par. 3.5, the criteria for selection of episodes (clean, medium pollution, and high pollution) are consistent with the discussion of clean, medium pollution, and high pollution days in Par. 3.4, i.e., average $PM_1$ concentration <20 μg m$^{-3}$ for clean episodes, 40 μg m$^{-3}$ < average $PM_1$ concentration <80 μg m$^{-3}$ for M-pollution episodes and average $PM_1$ concentration >100 μg m$^{-3}$ for H-pollution episodes. As shown in Figure 1, about 76% of the measurement periods have been grouped for episode analysis, except a few narrow time windows and the emissions-controlled parade period (early September 2015). For different episodes within the same pollution category, the sources and atmospheric processes of aerosol could be largely different in terms of the seasonality and meteorological conditions. Therefore, we divided the time series into different episodes to better investigate the factors determining the difference in different episodes. Regarding episode C1 and M1 in late summer and M7 in early winter, we only selected one episode to compare with others, because C1 (162 hrs) and M7 (212 hrs) have long time coverage and therefore are representative, while there was only one medium pollution episode (M1, 68 hrs) in late summer during our measurement period.

In the revised manuscript from page 12, line 35 to page 13, line 1, we have now added the following:

"To get a better insight into......seven clean episodes (average $PM_1$ concentration < 20 μg m$^{-3}$), seven M-pollution episodes (40 μg m$^{-3}$ < average $PM_1$ concentration < 80 μg m$^{-3}$) and five H-pollution episodes (average $PM_1$ concentration > 100 μg m$^{-3}$) were selected for further analysis".

3.  The analysis of the SOA production routes is very interesting, but RH is certainly not the best tracer for aqueous phase processes. I invite the authors to make use of the aerosol Liquid Water Content (LWC) instead, which is a much better tool for this purpose. It is not difficult to calculate the aerosol LWC, based on simple models, once RH, T and aerosol chemical composition are available.

Response: We agree with the referee that aerosol liquid water content (LWC), instead of RH, is a better parameter for investigating the aqueous-phase process. As shown in the figure below, the fraction of SOA increased when LWC was > ~25-35 μg m$^{-3}$ in late summer,

> ~15 µg m⁻³ in autumn and early winter. The increase of SOA fraction is much more clear when using LWC than using RH.

[Figure]

In the revised manuscript from page 14, line 29 to page 15, line 16, we have now changed "The variations of OA composition as functions of RH ……when RH increased." to "During late summer, the LWC ranged from 2.1 µg m⁻³ to 53.6 µg m⁻³, both the mass concentrations of LSOA and RSOA increased as LWC increased when LWC was higher than ~25-35 µg m⁻³……. These variations indicated the promotion of aqueous-phase processes on the formation of SOA especially during autumn and early winter with higher LWC."

In page 2, lines 10-12 in abstract, we have also changed "our analyses suggest that photochemical oxidation dominated SOA formation during all three seasons,……" to "our analyses suggest that both photochemical oxidation and aqueous-phase processing played important roles in SOA formation during all three seasons,……".

And in page 16, lines 25-27 in conclusion, we have changed "……found that photochemical processing dominates SOA formation during all three seasons……." to "……found that both photochemical processing and aqueous-phase processing played important roles in SOA formation during all three seasons……."

Also, in page 6, lines 19-24, we added "2.4 Liquid water content……" in Experimental section.

Specific comments

1. P7.L13. "OA dominated PM1 mass in late summer and autumn, whereas inorganic species played a more important role in early winter". This sentence is not supported by the results: the difference in OA contribution between autumn and early winter is almost negligible (49 vs 46%). Fig. S3 shows clearly that OA is the dominant component in early winter as well.

Response: As shown in Fig.1f, the mass fraction of OA has a decreasing trend from summer

to early winter during our measurement. Meanwhile, as shown in Fig.S3, the mass fractions of OA were 64% during late summer, 49% during autumn and 46% during early winter, with corresponding contributions of 36%, 51% and 54% for inorganic species during late summer, autumn and early winter, respectively. We therefore concluded that OA played a dominant role during late summer. We agree with the referee that the difference in OA contribution between autumn and early winter is small. Therefore, in the revised manuscript, we changed the sentence "OA dominated $PM_1$ mass in late summer and autumn, whereas inorganic species played a more important role in early winter" to "OA dominated $PM_1$ mass in late summer. In autumn and early winter, however, the contribution of OA decreased and secondary inorganic aerosol increased to be equally important."

2. P7.L27. It is not properly temperature that drives the boundary layer evolution. A lower temperature is a consequence of the lower solar radiation reaching the surface, as it is a shallower boundary layer.

Response: Change made. In the revised manuscript, we changed "Due to lower temperature in early winter, the planetary boundary layer (PBL) height was relatively flat compared to that in autumn and late summer, thus the noon peak of OA was more evident in early winter" to "Due to the relatively flat planetary boundary layer (PBL) height related to stagnant meteorological conditions in early winter compared to that in autumn and late summer, the noon peak of OA was more evident in early winter".

3. P8.L15-16. Also PBL dynamics may have an effect on this.

Response: Change made. In the revised manuscript, it now reads "The nighttime concentrations are generally high (Fig. S4), likely due to increased diesel fleets which are allowed in urban Beijing only at night and the decrease of PBL during nighttime."

4. Figure 3a. Please change the colors of the plotted lines. It is hard to distinguish one line from the other.

Response: Change made. In the revised manuscript, we have now changed the color of LSOA from pink to violet (see below). The color of LSOA in other figures were also changed accordingly.

[Figure]

**Referee #2**

In this study, the authors performed four-month measurement of the chemical composition of PM1 in Beijing, with an Aerosol Chemical Speciation Monitor (ACMS). PMF analysis is applied to study sources of organic aerosol. The authors perform routine analysis to study the regional vs. local sources of PM1, seasonal variation of PM1, and gas-phase vs. aqueous-phase formation pathways of organics and sulfate. The analysis procedure has been widely used in the literature. There have been numerous studies with ACSM or AMS to study the same topic in the same area (as summarized in Table 1). My overall impression is that this manuscript fails to establish enough novelty to distinguish itself from previous studies. Thus, I would not recommend this manuscript for publication in its current state.

Response: We agree with the referee that there are an increasing number of ACSM or AMS studies in urban Beijing, as already summarized in Table 1. However, the causes of fine PM pollution in urban Beijing are still not fully understood, most likely due to the campaign-to-campaign difference in meteorological conditions, emissions, and atmospheric processes. Also, there are only very limited studies investigating the seasonal difference so far (Sun et al., 2015; Hu et al., 2016). In our study, we present three-season measurements and discuss the seasonal difference in aerosol sources and formation processes. In particular, based on robust data analyses, we evaluate the importance of primary v.s. secondary species and of photochemistry v.s. aqueous-phase processes, in clean and polluted conditions in the three investigated seasons. Therefore, we believe that our present study provides essential information to the scientific community to improve our understanding of fine PM pollution.

1. The discussions on regional SOA factor (RSOA) and local SOA factor (LSOA) suffers major flaws in its logical flow. To start with, it is not justified why these two OA factors are classified as regional and local. In section 3.3, the authors simply state that two oxygenated OA factors were identified and they are local SOA and regional SOA, but no

justification is provided. Moreover, the assumption that the one factor is regional and the other factor is local is applied throughout the manuscript to infer the sources of sulfate (P9 L25-35), OA (P11 L1-5), and PM1. However, this assumption is not justified yet! In P10 L1-15, bivariate polar plots of SO4 are used to discuss the sources of SO4. The results are reasonable. However, the sources of SO4 need to be established first and further used to infer the sources of LSOA and RSOA, instead of the other way around.

Response: We agree with the referee about the need for further justification in the discussion of regional and local SOA. As discussed above in "response to referee #1", we have now reconstructed this part to be more logical. We now first discuss the source regions of sulfate, then discuss the correlation of these two SOA factors with sulfate, and then their mass spectra (e.g., $f_{44/43}$). To further support the distinction of RSOA and LSOA, we have now included the potential source regions of LSOA and RSOA. The polar plots below show clearly that LSOA is mainly located in the sampling site while RSOA is mainly from the south to the sampling site.

[Figure]

[Figure]

In the revised manuscript from page 9, line 25 to page 10, line 15, we have now updated and reconstructed this part by discussing sulfate before SOA. It now reads "In order to analyze sources of sulfate in our study period, the bivariate polar plots of sulfate during different seasons are displayed in Fig. 3. ……… These results indicate that transported sulfate at a large regional scale was more important during late summer, while local formation was the major source of sulfate in early winter due to residential heating. Two oxygenated OA factors with much different time series were identified in our study which we defined as local SOA (LSOA) and regional SOA (RSOA) as characterized below in details…….. different correlations between sulfate and RSOA or LSOA were found during different seasons……… suggested that RSOA is related to regional source of OOA and LSOA indicates local sources and subsequent local formation."

In page 10, lines 22-24, we also added "……than locally formed SOA (Sun et al., 2014, 2015). The attribution of LSOA and RSOA is further supported by the bivariate polar plots (Fig. S5), which show clearly that LSOA is mainly located in the sampling site while RSOA is mainly from the south to the sampling site."

From page 1, line 30 to page 2, line 4, we have also changed "Distinctly different

correlations between RSOA and sulfate were found in our study…… while local and/or nearby sulfate formation may be more important in winter" to "The sulfate source regions analysis implies that sulfate was mainly …… Meanwhile, distinctly different correlations between sulfate and RSOA or LSOA……confirmed the regional characteristic of RSOA and local property of LSOA" in abstract.

2. I also have concerns on the SO4 discussions. (1) It is important to note that particle liquid water content is a better proxy for aqueous phase reaction than RH. It has been well-established that the particle LWC not only depends on RH, but also on particle composition. (2) Figure 3a suggests that regional transport is the major source of SO4 in later summer. However, the good correlation between SO4 and Ox in figure 6 suggests SO4 may be locally formed. How to reconcile this difference?

Response: We fully agree with the referee that aerosol LWC is a better proxy for aqueous-phase reaction. However, when discussing the sulfur oxidation ratio ($F_{SO4}$), it is usually presented as a function of RH as shown in many previous studies (e.g., Sun et al., 2013, 2014; Yang et al., 2015; Elser et al., 2016; Li et al., 2017). Consistent with previous studies, the sulfur oxidation ratio ($F_{SO4}$) increases exponentially when RH is larger than ~50% during winter, suggesting the efficient formation from aqueous-phase processes during winter. Following the referee's suggestion, we also plotted $F_{SO4}$ against LWC but did not find clear evidence.

Regarding the results in Figure 3a and Figure 6, we think that they are consistent. As discussed in the book "Atmospheric Chemistry and Physics" (Seinfeld and Pandis, 2016) and a review paper in Chemical Reviews (Zhang et al., 2015), at the typical atmospheric level of OH radical, the lifetime of $SO_2$ from the reaction with OH is about 1 week. Thus, $SO_2$ oxidation into sulfate may proceed during long-range transport (Rodhe et al., 1981). Our results in Figure 6 show that sulfate is mainly formed from photochemical oxidation, which may happen during regional transport, consistent with result in Figure 3a.

In the revised manuscript page 14, lines 23-26, we have now added the following discussion:

"It should be noted that at the typical atmospheric level of OH radical, the lifetime of $SO_2$ from the reaction with OH is about 1 week (Seinfeld and Pandis, 2016; Zhang et al., 2015). Thus, $SO_2$ oxidation into sulfate may proceed during long-range transport in late summer (Rodhe et al., 1981), consistent with our results in Figure 3."

Reference:

Elser, M., Huang, R. J., Wolf, R., Slowik, J. G., Wang, Q., Canonaco, F., Li, G., Bozzetti, C., Daellenbach, K. R., Huang, Y., Zhang, R., Li, Z., Cao, J., Baltensperger, U., El-Haddad, I., and Prévôt, A. S. H.: New insights into PM2.5 chemical composition and sources in two major cities in China during extreme haze events using aerosol mass spectrometry, Atmos. Chem. Phys., 16, 3207–3225, 2016.

Li, H., Zhang, Q., Zhang, Q., Chen, C., Wang, L., Wei, Z., Zhou, S., Parworth, C., Zheng, B., Canonaco, F., Prévôt, A. S. H., Chen, P., Zhang, H., Wallington, T. J., and He, K.: Wintertime aerosol chemistry and haze evolution in an extremely polluted city of the North China Plain: significant contribution from coal and biomass combustion, Atmos. Chem. Phys., 17(7), 4751-4768, 2017.

Rodhe, H., Crutzen, P., and Vanderpol, A.: Formation of Sulfuric and Nitric-Acid in the Atmosphere during Long-Range Transport, Tellus, 33, 132–141, 1981.

Seinfeld, J. H., Pandis, S. N.: Atmospheric chemistry and physics: from air pollution to climate change, John Wiley & Sons, 2016.

Sun, Y., Wang, Z., Fu, P., Jiang, Q., Yang, T., Li, J., and Ge, X. L.: The impact of relative humidity on aerosol composition and evolution processes during wintertime in Beijing, China, Atmos. Environ., 77, 927-934, 2013.

Sun, Y., Jiang, Q., Wang, Z., Fu, P., Li, J., Yang, T., and Yin, Y.: Investigation of the sources and evolution processes of severe haze pollution in Beijing in January 2013, J. Geophys. Res. Atmos., 119, 4380-4398, 2014.

Yang, Y. R., Liu, X. G., Qu, Y., An, J. L., Jiang, R., Zhang, Y. H., Sun, Y. L., Wu, Z. J., Zhang, F., Xu, W. Q., and Ma, Q. X.: Characteristics and formation mechanism of continuous hazes in China: a case study during the autumn of 2014 in the North China Plain, Atmos. Chem. and Phys., 15(14), 8165, 2015.

Zhang, R. Y., Wang, G. H., Guo, S., Zamora, M. L., Ying, Q., Lin, Y., Wang, W. G., Hu, M., and Wang, Y.: Formation of urban fine particulate matter, Chem. Rev., 115(10), 3803-3855, 2015.

---

## Author Comment (AC3) · 8 Jun 2019

**Supplement of:**

[revised manuscript text omitted]

---

## Editor Decision (ED1)

Dear Dr. Huang,

Thank you very much for responding to the comments from the two referees and for revising the manuscript.

I have sent the revised manuscript to the two original referees for review, as they requested. While both referees stated that the revised manuscript has improved substantially, one of the referees mentioned that the concerns about the sulfate issue remain, and would like the authors to address this. Please see the referee's comments pasted below:

"I thank the authors for addressing my comments. The manuscript is largely improved. However, my concerns on the sulfate discussions remain.

The authors stated that they fully agree with that aerosol LWC is a better proxy for aqueous-phase reaction. Therefore, RH must be fully replaced by aerosol LWC throughout the manuscript when probing the role of aqueous-phase reaction. However, the authors still use RH as proxy, just because previous studies usually plot FSO4 vs RH! A method usually used in previous studies is not guaranteed the method is right.

The authors mentioned that there is no clear evidence between FSO4 and LWC, indicating limited role of aqueous-phase. To me, this is an important conclusion - A conclusion that challenges "well-established conclusion" from previous studies based on RH.

In figure 6 (d)(e)(f), data from different seasons should be presented in a consistent way. For example, in (d) and (f), data should be grouped by RH > / < 65% and then fitted. I am requesting this because FSO4 seems to have some correlation with Ox when RH<65% in panel (f) by eyeballing.

Lastly, to support the argument that sulfate is formed from photooxidation during regional transport, please show prove that Ox is regional as well (i.e., please show the bivariate polar plots of Ox of late summer)."

Please respond to the referee's comments and submit a revised manuscript, as appropriate.

Sincerely,

Luisa Molina

---

## Author Response (AR2)

The authors thank the editor and referee to review our manuscript and particularly for the valuable comments and suggestions that have significantly improved the manuscript. We provide below point-by-point responses to the referee' comments. We also have made most of the changes suggested by the referee in the revised manuscript.

Referee #2

I thank the authors for addressing my comments. The manuscript is largely improved. However, my concerns on the sulfate discussions remain.

The authors stated that they fully agree with that aerosol LWC is a better proxy for aqueous-phase reaction. Therefore, RH must be fully replaced by aerosol LWC throughout the manuscript when probing the role of aqueous-phase reaction. However, the authors still use RH as proxy, just because previous studies usually plot FSO4 vs RH! A method usually used in previous studies is not guaranteed the method is right.

The authors mentioned that there is no clear evidence between FSO4 and LWC, indicating limited role of aqueous-phase. To me, this is an important conclusion - A conclusion that challenges "well-established conclusion" from previous studies based on RH.

In figure 6 (d)(e)(f), data from different seasons should be presented in a consistent way. For example, in (d) and (f), data should be grouped by RH > / < 65% and then fitted. I am requesting this because FSO4 seems to have some correlation with Ox when RH<65% in panel (f) by eyeballing.

**Response:** We thank the referee's suggestions on the sulfate formation discussion. We have now replaced RH using aerosol LWC (ALWC) to probe the role of aqueous-phase reaction in the revised manuscript. Meanwhile, according to the referee's suggestion, we also grouped data by RH > / < 65% and then fitted for late summer and early winter. As shown in the figures below, during late summer, positive correlations between $F_{SO4}$ and $O_x$ with similar slopes and correlation coefficients in RH < 65% and RH > 65% were observed, suggesting the important role of photochemical oxidation during late summer irrespective of the RH range. During early winter, however, at RH < 65% sulfate was also formed by photochemical oxidation because of the positive correlation between $F_{SO4}$ and $O_x$. The relationships between $F_{SO4}$ and ALWC during different seasons at RH > 65% and low atmospheric oxidative capacity ($O_x$ < 60 ppb) were investigated. There were positive correlations between $F_{SO4}$ and ALWC during all three seasons, indicating the contribution of aqueous-phase processing to the sulfate formation in high RH condition. Meanwhile, we found that during late summer $F_{SO4}$ was up to ~0.6 with $O_x$ while only up to ~0.3 with ALWC, suggesting the more important role of photochemical oxidation in the sulfate formation during late summer. On the contrary, during early winter the increase of $F_{SO4}$ with ALWC (from ~0.05 to ~0.5) was more efficient than that with $O_x$ (from ~0.05 to ~0.2), indicating that aqueous-phase reactions were more responsible during early winter. During autumn, $F_{SO4}$ was up to about 0.4-0.5 both with $O_x$ and ALWC, suggesting that for sulfate formation both photochemical oxidation and aqueous-phase reactions had important contributions during autumn.

[Figure]

In the revised manuscript page 13, lines 3-22, we have now changed the sulfate discussion "Fig. 6a-c plots $F_{SO4}$ versus RH, colored by $O_x$ (= $O_3$ + $NO_2$) concentration......The relationship between $F_{SO4}$ and $O_x$ during different seasons are also shown in Fig. 6d-f......These results suggest ...... in autumn and early winter" to "Fig. 6a-c plots $F_{SO4}$ versus $O_x$ (= $O_3$ + $NO_2$) concentration which is a tracer to indicate photochemical processing during late summer, autumn and early winter, respectively. During late summer, positive correlations between $F_{SO4}$ and $O_x$ with similar slopes and correlation coefficients in RH < 65% and RH > 65% were observed, suggesting the important role of photochemical oxidation during late summer irrespective of the RH range. During autumn and early winter, at RH < 65% sulfate was also formed by photochemical oxidation because of the positive correlations between $F_{SO4}$ and $O_x$, while there was no correlation between $F_{SO4}$ and $O_x$ at RH > 65%, indicating that other processes (e.g., aqueous-phase reactions) may contribute to the sulfate formation in high RH condition. This is supported by the relationships between $F_{SO4}$ and ALWC at RH > 65% and low atmospheric oxidative capacities of $O_x$ < 60 ppb (Fig. 6d-f). There were positive correlations between $F_{SO4}$ and ALWC during all three seasons in high RH condition, indicating the contribution of aqueous-phase processing to the sulfate formation. Meanwhile, we found that $F_{SO4}$ was up to ~0.6 with $O_x$ while only up to ~0.3 with ALWC during late summer, suggesting the more important role of photochemical oxidation on the sulfate formation during late summer. On the contrary, during early winter the increase of $F_{SO4}$ with ALWC (from ~0.05 to ~0.5) was more efficient than that with $O_x$ (from ~0.05 to ~0.2), indicating that aqueous-phase reactions were more responsible during early winter. During autumn, $F_{SO4}$ was up to about 0.4-0.5 both with $O_x$ and ALWC, suggesting that for sulfate formation during autumn both photochemical oxidation and aqueous-phase reaction had important contributions".

In page 2, lines 10-12 in abstract, we have also changed "while for sulfate formation, gas-phase photochemical oxidation was the major pathway in late summer and heterogeneous processes were likely more important in autumn and early winter" to "while for sulfate formation, gas-phase photochemical oxidation was the major pathway

in late summer, aqueous-phase reactions were more responsible during early winter and both processes had contributions during autumn".

And in page 15, lines 6-8 in conclusion, we have changed "gas-phase photochemical oxidation was the major formation mechanism of sulfate in late summer, while aqueous-phase chemistry was likely playing an important role in autumn and early winter" to "for sulfate formation, both photochemical oxidation and aqueous-phase reaction had contributions during autumn, while photooxidation played a more important role during late summer and aqueous-phase reactions were more responsible during early winter".

Lastly, to support the argument that sulfate is formed from photooxidation during regional transport, please show prove that Ox is regional as well (i.e., please show the bivariate polar plots of Ox of late summer).

**Response:** We thank the referee's suggestion and have added the bivariate polar plots of $O_x$ of late summer in the supplementary information of the revised manuscript to support our conclusion. As shown in the figure below, high $O_x$ concentration mainly located in the south region from the sampling site, further indicating that sulfate is formed from photooxidation during regional transport.

[Figure]

Ox (ppb)

[revised manuscript text omitted]